# Protocol for a feasibility study of smoking cessation in the surgical pathway before major lung surgery: Project MURRAY

Sebastian T Lugg [ID],[1] Amy Kerr,[2] Salma Kadiri,[2] Alina-Maria Budacan,[2] Amanda Farley,[3] Olga Perski,[4] Robert West,[4] Jamie Brown,[4] David R Thickett,[1] Babu Naidu,[1,2] on behalf of the Project MURRAY Investigators

► Prepublication history and supplemental material for this paper are available online. To view these files, please visit the journal online (http://dx.doi.org/10.1136/bmjopen-2019-036568).

[1]Birmingham Acute Care Research Group, Institute of Inflammation and Ageing, University of Birmingham, Birmingham, UK
[2]Department of Thoracic Surgery, University Hospitals Birmingham NHS Foundation Trust, Birmingham, UK
[3]Insitute of Applied Health Research, University of Birmingham, Birmingham, UK
[4]Institute of Epidemiology & Health, University College London, London, UK

**Correspondence to**
Babu Naidu;
b.naidu@bham.ac.uk

## ABSTRACT

**Introduction** Smoking prior to major thoracic surgery is the biggest risk factor for development of postoperative pulmonary complications, with one in five patients continuing to smoke before surgery. Current guidance is that all patients should stop smoking before elective surgery yet very few are offered specialist smoking cessation support. Patients would prefer support within the thoracic surgical pathway. No study has addressed the effectiveness of such an intervention in this setting on cessation. The overall aim is to determine in patients who undergo major elective thoracic surgery whether an intervention integrated (INT) into the surgical pathway improves smoking cessation rates compared with usual care (UC) of standard community/hospital based NHS smoking support. This pilot study will evaluate feasibility of a substantive trial.

**Methods and analysis** Project MURRAY is a trial comparing the effectiveness of INT and UC on smoking cessation. INT is pharmacotherapy and a hybrid of behavioural support delivered by the trained healthcare practitioners (HCPs) in the thoracic surgical pathway and a complimentary web-based application. This pilot study will evaluate the feasibility of a substantive trial and study processes in five adult thoracic centres including the University Hospitals Birmingham NHS Foundation Trust. The primary objective is to establish the proportion of those eligible who agree to participate. Secondary objectives include evaluation of study processes. Analyses of feasibility and patient-reported outcomes will take the form of simple descriptive statistics and where appropriate, point estimates of effects sizes and associated 95% CIs.

**Ethics and dissemination** The study has obtained ethical approval from NHS Research Ethics Committee (REC number 19/WM/0097). Dissemination plan includes informing patients and HCPs; engaging multidisciplinary professionals to support a proposal of a definitive trial and submission for a full application dependent on the success of the study.

**Trial registration number** NCT04190966.

## Strengths and limitations of this study

► This study addresses smoking in major thoracic surgery which can result in a significant economic and healthcare burden.
► This study will assess the role of a hybrid model of integrated smoking cessation within the surgical pathway delivered by trained healthcare practitioners and use of a web-based application.
► This feasibility study will assess patient recruitment to inform a definitive study.
► This study will add to the limited evidence towards of effective smoking cessation strategies in major thoracic surgery.
► This feasibility study will not answer the overarching research question of efficacy but will directly inform a well-designed definitive study.

the risk of developing postoperative pulmonary complications including pneumonia and lung collapse. Just under half of smokers develop these complications with associated increased in-hospital mortality (0.5%–12%), intensive therapy unit admissions (1.5%–26%), increased hospital stay (5–14 days) and poorer long-term outcomes.[2–4] Furthermore, lung cancer surgery patients have an 86% increased risk for cancer recurrence and twofold decrease in 5 year survival compared with patients who quit smoking at diagnosis.[5]

### The need for integrated hospital-based smoking cessation support

The current National Institute of Health and Care Excellence (NICE) guidance is that smokers undergoing 'elective' surgery should receive behavioural support and stop-smoking pharmacotherapy as early as possible in their outpatient or preoperative assessments. This should be offered weekly, preferably face-to-face, for a minimum of 4 weeks after the quit date.[6] The Cochrane review supporting this

## INTRODUCTION
### Poorer outcomes for smokers in thoracic surgery

25 000 patients undergo major thoracic surgery every year in the UK.[1] One in five patients smoke before surgery, which increases

type of 'intensive' intervention[7] was based on two small trials in orthopaedics and general surgery.[8 9]

However, in current practice, most thoracic surgery patients do not receive any preoperative smoking cessation support. Of the 120 patients attending a large UK thoracic regional unit, only 40% of current smokers were offered support,[10] similar to the proportion of all eligible preoperative patients agreeing to participate in smoking cessation studies.[11 12] Only one in three smokers self-report abstinence at the time of lung cancer surgery,[4] with biochemical verification indicating much lower quit rates.[7]

Thus, the current usual care (UC) of referral to stop-smoking services does not meet the standard set by NICE. This may be due to these services being designed to promote long-term quitting, which many smokers undergoing surgery may not be willing to commit to. Many patients also report that attending smoking cessation clinic appointments during their work-up for surgery is a significant barrier to stopping smoking and would prefer bespoke support during hospital visits.[13]

In preoperative smoking cessation studies, behavioural support is delivered either by research nurses or independent smoking cessation practitioners.[12 14] It is logical that healthcare practitioners (HCPs) in the surgical pathway could be trained to deliver the support due to the high prevalence of smoking within this patient group. This approach is advocated by both the Lung Cancer Nurse Forum and the Society of Cardiothoracic Surgery (UK) and evidence suggests that nurse delivered smoking cessation interventions are effective.[15] Timely access is also crucial; NICE lung cancer guidelines recommend that surgery should not be delayed to give up smoking[16] and needs to be performed within 62 days of presentation and 31 days of diagnosis to avoid heavy financial penalties on individual Trusts.[17] Therefore, avoiding delays in receiving behavioural and pharmacological support is of paramount importance.

### Poor quality of available smoking cessation applications

A combination of a web-based application (web app) with structured face-to-face behavioural support may aid successful quit attempts, while also supporting HCPs in delivering smoking cessation support throughout the surgical pathway. A recent review of over 100 smoking cessation apps showed that only six were deemed to be of high quality.[18] There are no apps specifically designed for smoking cessation in patients undergoing major surgery and none designed to provide hybrid support alongside a trained smoking cessation practitioner.

### Summary

Many patients who undergo thoracic surgery continue to smoke, and effective preoperative smoking cessation interventions may improve outcomes. However, there are few studies exploring strategies for integrating smoking cessation into the patient pathway, which is a high priority area for research in these patients. We have developed a bespoke, tailored intense, integrated smoking cessation intervention to test in a 'real-life' clinical trial within the UK. The intervention involves pharmacotherapy and a hybrid of support delivered by trained HCPs in the thoracic surgical pathway and a complementary web app. Support may be particularly effective because of site of the surgery, the lungs, makes this a clear 'teachable moment' for patients.[19 20]

### Study aim

The overall aim of this research is to determine if integrated (INT) smoking cessation support in the surgical pathway improves smoking cessation rates in patients undergoing major elective thoracic surgery when compared with UC of standard community/hospital-based NHS smoking cessation support. To answer this research question with substantial evidence of the clinical and cost-effectiveness of INT approach, a multicentre randomised controlled trial (RCT) is required. Feasibility studies are a recommended prerequisite to assess feasibility of a large and expensive full-scale trial. We have therefore designed this multicentre feasibility study, which aims to evaluate the feasibility of the INT by making the following quantitative and qualitative assessments.

### Objectives of the feasibility study

The aims of the feasibility study are to assess various aspects of the trial design and management and not to determine the relative effectiveness of INT versus UC.

### Primary objective

To establish the number of patients who agree to participate in the intervention as a proportion of those eligible to enter the study.

### Secondary objectives

1. Integration of the intervention into the clinical pathway by time from decision to operate from study recruitment.
2. Explore barriers to study recruitment, including descriptive reasons for non-participation from screening logs.
3. Fine-tune study procedures and pilot data capture forms aiming for over 90% completion of important perioperative data for each patient.
4. To assess the proportion of patients in the INT group who have quit smoking by the day of surgery and 1 month after surgery.
5. To assess the proportion of patients in the observation only UC group who have quit smoking by the day of surgery and 1 month after surgery.
6. To define the variability of smoking cessation practices in all patients using the nicotine replacement usage questionnaire.
7. Qualitative interview: to understand patients' experiences of and engagement with the intervention, and any unintended consequences; to establish whether the intervention is acceptable to thoracic surgery patients and staff and investigate recommendations for optimisation of intervention delivery.

## TRIAL DESIGN

### Design

Project MURRAY is an RCT comparing the effectiveness of INT verses UC in smoking cessation rates in patients undergoing major thoracic surgery, taking form of a stepped wedge cluster randomised controlled trial,[21] and as such will not require individual patient randomisation. This feasibility study will evaluate the substantive trial and study processes.

### Setting

Trial recruitment will be over a period of 12 months with an additional 6-month follow-up period. Recruitment will initiate at the University Hospitals Birmingham NHS Foundation Trust, which is the trial co-ordinating site and performs >1000 major thoracic surgical procedures a year. Additional recruitment will involve four further regional thoracic surgical centres performing >400 major thoracic surgical procedures a year.

### Flow of participants during the trial

The anticipated pathway of patients through the trial is shown in the trial schema (figure 1). All adult patients who fulfil the inclusion and exclusion criteria during the study period will be approached and participant information sheets (PIS) will be provided. Written informed consent will be obtained after an opportunity for patients to discuss requirements for the study. If the patient accepts the intervention, they will be placed into the INT group of the study. If they decline the intervention, they will be invited to take part in the observational only part of the study and will receive UC. We will extract routinely

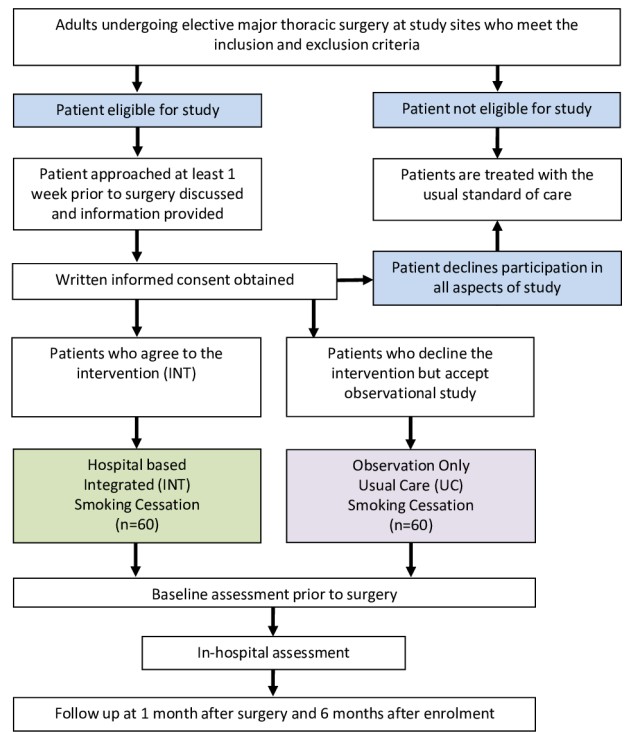

**Figure 1** Trial schema. INT, intervention integrated; UC, usual care.

collected preoperative and postoperative data from patient's medical records and also collect data using questionnaires. Adverse events will be collected throughout the duration of the study. The summary of assessments is detailed in (table 1).

### Study eligibility

#### Inclusion criteria

► Current tobacco smoker (smoked within the last 28 days).
► Major thoracic surgery (including both open and minimally invasive approach).
► Able to provide written informed consent.
► At least 1 weeks' time to surgery.
► Age over 18 years.

#### Exclusion criteria

► Emergency thoracic surgery.
► Inability to perform exhaled carbon monoxide (CO) measurements.

### Patient identification and screening procedure

Patients who are listed for major thoracic surgery will be identified and screened for eligibility prior to surgery. If a patient is screened and not eligible for the study or does not consent to be in the intervention or observation group, a record of the case will be kept in the screening log and will inform recruitment targets. No further information will be collected on ineligible patients or those that have not given consent for participation in the study.

### Patient recruitment

The PIS, developed with feedback from our Patient and Public Involvement (PPI) representatives, will be sent to the patient before the initial consultation, allowing time for the patient to review and ask questions. As part of the normal consultation process, the HCP will deliver a brief intervention detailing the importance of smoking cessation as per standard clinical practise (National Centre for Smoking Cessation Training (NCSCT)—Ask, Advice, Act). Patients will then be asked to consider and consent to a trial testing the new bespoke integrated smoking cessation support delivered in secondary care by the surgical/nursing team. If a patient does not wish to enter the INT arm, they will be approached to take part in the UC arm of the study.

If patients want to join the trial on the day of the clinic appointment, they can do so, as the intervention is low risk and aimed at being offered 'there and then'. A research team member will obtain written informed consent with delegated authority from the Principal Investigator (see online supplemental participant consent form). A copy of the signed consent form will be given to the participants and a copy will be placed in the medical notes. The original consent form will be stored in the investigators site file. Consent will be sought at every study contact and participants will be made aware that they are free to withdraw consent at any time without reprisal. Participants will be

**Table 1** Summary of investigations and assessments

| | Participants | | Time points | | | | | |
|---|---|---|---|---|---|---|---|---|
| | Intervention (INT) group | Usual care (UC) group | Baseline clinic prior to surgery | Day of surgery | During hospital stay | Day of hospital discharge | 1 month after surgery | 6 months after enrolment |
| Eligibility and written informed consent | X | X | X | | | | | |
| Demographic data* | X | X | X | | | | | |
| Previous medical history† | X | X | X | | | | | |
| Self-reported quit rate | X | X | X | X | | X | X | X |
| Exhaled CO measurement | X | X | X | X | | X | X | X |
| NRT and support usage questionnaire | X | X | X | X | | X | X | X |
| Fagerstrom test for nicotine dependence | X | X | X | | | | X | |
| Mood and physical symptoms scale | X | | X | X | | X | X | |
| EQ-5D-5L | X | | X | | | | X | |
| Health resource usage questionnaire | X | | | | | | X | |
| Surgery and anaesthetic data‡ | X | X | | X | | | | |
| Postoperative complications§ ¶ | X | X | | | X | X | X | |
| Hospital readmission** | X | X | | | | | X | |
| Semistructured qualitative interviews†† | X | X | | | | | X | |
| Adverse events | If applicable | | | | | | | |
| Protocol deviations | If applicable | | | | | | | |

*Demographic data: gender, age, indication for surgery, height, weight, BMI, ASA grade, ECOG score, dyspnoea score, recent lung function.
†Previous medical history: smoking history, alcohol intake per week, comorbidities (COPD, Ischaemic Heart Disease, Congestive Cardiac Failure, Hypertension, diabetes (diet-controlled/oral therapy/insulin), renal failure, previous stroke, thyroid disease (hyperthyroid/ hypothyroid).
‡Operation performed (side of surgery, operation, surgical technique).
§Postoperative data and observations: routine blood results if done (full blood count, albumin, renal function, electrolytes, CRP). Acute complications: according to ESTS[30] (see online supplemental appendix A) and Thoracic Morbidity and Mortality Classification[31] (see online supplemental appendix B), data also collected including admission and length of stay on the ward (0), step-down,[1] the HDU[2] and ITU.[3] Data will also be collected in patients requiring mini-tracheostomy or additional surgery. Postoperative pulmonary complications: using stEP-COMPAC Group definition of postoperative pulmonary complications[32] (see online supplemental appendix C) defining atelectasis (detected on computer tomography/CXR), pneumonia (using US Centres for Disease Control criteria), acute respiratory distress syndrome (using Berlin Consensus), and pulmonary aspiration (clear clinical history and radiological evidence).
¶Discharge data: total hospital stay, home with flutter bag, histology data and mortality.
**Follow-up: hospital readmission up to and including 1 month following surgery.
††At 4–8 weeks postsurgery patients will also have semistructured qualitative patient interviews will be undertaken at 4 weeks postdischarge to investigate experience, engagement, acceptability, unintended consequences/benefits and how to optimise the intervention delivery.
ASA, American Society of Anethesiologist; BMI, body mass index; CO, carbon monoxide; COPD, chronic obstructive pulmonary disease; CRP, C-reactive protein; ECOG, Eastern Cooperative Oncology Group; ESTS, European Society of Thoracic Surgery; HDU, high-dependency unit; INT, intervention integrated; ITU, intensive therapy unit; NRT, nicotine replacement therapy; UC, usual care.

consented to inform their GP of their involvement in the study.

### Intervention development

A smoking cessation package has been developed using best practice from NICE/NCSCT involving pharmacotherapy and a hybrid of behavioural support delivered by trained HCPs in the thoracic surgical pathway and a complementary web app. *Quit4Surgery* is a web app created using a user-centred design approach and developed through a series of design workshops with patients, HCPs, academic researchers with expertise in smoking cessation and informed by behavioural frameworks/motivation theory. The iterative development was handled with 'Agile' and Scrum project management.[22 23] *Quit4Surgery* contains behaviour change techniques that research suggests can improve the chances of quitting, including

goal setting, self-monitoring, feedback, rewards, information about health consequences, advice on medication use, advice on changing routines, advice on coping and support for identity change.[24] *Quit4Surgery* also enables HCPs to record and enter patients' CO levels.

### Integration of full package of support into the surgical pathway (INT)

1. INT will be delivered by key HCPs in the surgical pathway (ie, surgical nurses, lung cancer nurses or preoperative assessment nurses) who have received the NCSCT training to deliver behavioural support required for stop smoking practitioners with specific focus on major surgical patients.[25] The initial consultation of 15–30 min will occur either on the day of consent or at the earliest convenience for the patient. This will be in a private room and will outline the benefits of stopping

 Lugg ST, *et al. BMJ Open* 2020;**10**:e036568. doi:10.1136/bmjopen-2019-036568

smoking before thoracic surgery, discuss pharmaco-therapy options, provide behavioural support,[26–29] decide on an early quit date (aim within 48 hours of consent) and provide further information regarding *Quit4Surgery*. The HCP will validate and quantify smoking amount using CO measurements, which will be repeated at subsequent face-to-face interactions, providing biofeedback on the success of smoking cessation to the patient.

2. Pharmacotherapy is encouraged for all patients and includes either combined short-acting and long-acting nicotine replacement therapy (NRT) or varenicline, which are provided as standard care within the NHS. Patients wishing to use e-cigarettes will be given advice as per the NCSCT guidance.[30] Treatment will be maintained for the peri-operative period to offset nicotine withdrawal symptoms and is typically 8–12 weeks for NRT and 12 weeks for varenicline as per NCSCT guidance, with pharmacotherapy and behavioural support tailored appropriately to individual patients.

3. *Quit4Surgery* will collect feedback regarding the patient's engagement, cravings and abstinence and provide motivational feedback to support the patient. The motivational feedback provided by the web app is guided by established behavioural change theory and will complement the patients smoking cessation package. Patient feedback using the web app to the surgical team will guide them as to the need for additional contact if the patient wishes. This will help improve overall efficacy of the intervention.

INT patients will receive proactive support within 2 days of the quit date (in person or by telephone) and then weekly until 1 month after surgery. The weekly sessions will concentrate on facilitators and barriers to quitting, relapse, pharmacotherapy side effects and withdrawal symptoms. Face-to-face interactions will occur during the surgical outpatient appointments, preclerking clinic and during hospital admission to reduce the number of additional visits. If patients are unable to attend face-to-face visits, videoconferencing or telephone visits will be attempted. Thus, the INT will fit into the 'referral to treatment' target time frame and ensuring surgery is not delayed as recommended by NICE.

## Control group of UC

As part of the feasibility study, for those patients who decline consent to receive the intervention, we ask if they would consent to be observed during the thoracic surgical pathway. Termed UC, per usual local practice, patients are typically given a leaflet about the benefits of smoking cessation and referred to their local NHS smoking cessation service, which typically last up to 12 weeks and include pharmacotherapy. Patients will also be invited for an optional telephone interview after discharge. This is to help understand smoking cessation rates in UC as well as the reasoning for non-participation in the intervention. Both groups will receive the same preoperative, perioperative and postoperative care as per protocol.

## Withdrawal from the trial

Withdrawal from the trial before surgery is a decision of the participant. However, participants will be asked if the research team can still use the data collected during their participation in the research analyses.

## Protocol deviations

All study and protocol deviations will be documented in the patients case report form (CRF) and reported to the Principle Investigator, who will notify the Sponsor of any serious breaches. Patients will be analysed according to group allocation, by intent-to-treat analysis.

## Patient and public involvement

The project details were discussed at a national thoracic surgery patient group ('RESOLVE') meeting and feedback regarding merit and acceptability of the proposed intervention were incorporated. Dissemination of results will occur via specific patient feedback events. A patient representative was an active contributor to the development of the trial and intervention and will be a member of the trial management group. The intervention including the web app *Quit4Surgery* were designed to meet patients' specific 'needs' and 'drivers' of smokers who are waiting for thoracic surgery, with a PPI group involved in the design and initial product testing stages of this feasibility study.

## OUTCOME AND DATA COLLECTION
### Patient recruitment into study

The overall aims of the feasibility are to find out if a larger trial is feasible. The quantitative measurements related to this include the following:

▶ Proportion of all elective thoracic procedures screened.
▶ Proportion of eligible participants of those screened.
▶ Proportion of eligible participants consented to receive intervention.

In this feasibility study, selecting five units whose overall elective thoracic patient throughput amounts to 4000 patients a year, of whom 20% are smokers and 20% agree to take part in the study. It is expected that 60 eligible patients will be recruited to the INT group and 60 patients to the UC group of the study, with recruitment of 120 patients in total. Therefore, the aim is to recruit five patients a month to each group over the 12 months recruitment period across all sites.

### Patient identification and screening

The proportion of patients screened for eligibility and recorded on a screening log will be assessed and reported as the proportion of patients screened from the total number of planned major thoracic surgery during the study period.

### Reasons for failure to be recruited

The proportion of patients who were missed, which should be minimal and proportion of patients who

decline to take part will be recorded. Patients decline participation for many reasons, which should be captured whenever possible.

### Education material of nurses and surgeons
Feedback on the appropriateness, value and acceptability of the training will be elicited to enable refinement of the training programme for the substantive study, and to define a minimum competence. The training material will be evaluated for its ease of use, should it be used in the substantive study.

### Assessment of data collection process
Data will be collected using a CRF and will include demographic information and comorbidities. Postoperative complications will be defined by the European Society of Thoracic Surgery,[31] Thoracic Morbidity and Mortality system[32] and the stEP-COMPAC Group[33] (see online supplemental appendix for definitions). Hospital readmission rate will be determined within 30 days of discharge.

Assessment and identification will be made for loss of data during in-hospital stay to improve the data collection process for the substantive trial.

### Smoking-related outcomes
In this study, the feasibility of the following questionnaires will be tested:
► Self-reported quit rate and exhaled CO measurement.
► Fagerstrom Test for Nicotine Dependency: assessment of nicotine addiction.[34]
► Mood and Physical Symptoms Scale: assessment of cigarette withdrawal symptoms over the past 24 hours, including the strength of urge to smoke.[35]
► Generic health-related quality of life (EQ-5D-5L): assessment to provide a preference-based measure of health-related quality-of-life.[36]
► NRT and support usage questionnaire.
► Health resource usage questionnaire: assessment of resource will be assessed via patient-recall with resources being measured including planned hospital overnight stays, planned hospital outpatient visits, hospital emergency visits, hospital admissions, GP and other community service visits.

### Acceptability to and impact on patients
All patients consenting to participate in the trial will be eligible for interview and selected using maximum variety sampling by age, sex, ethnicity, admitting diagnosis, surgical procedure and smoking status. Interviews will be conducted until saturation is achieved, which is likely to be around 25–30 patients across all sites.[37] Telephone interviews will be conducted as to minimise impact to patients following major surgery, will last no longer than 60 min and will be audio recorded and transcribed by the researcher.

An interview guide will be developed using evidence from previous experience of running the rehabilitation and pain study, and based on the interview objectives: presurgical and postsurgical experiences of patients receiving the intervention (including effectiveness of staff communication) and patient engagement with it, any unintended consequences, acceptability of the intervention to patients and recommendations for how its use or content/design could be improved.

### Assessment of trial processes and impact on staff
Key HCPs in the clinical pathway will be invited to attend a focus group or individual interviews that will explore both the acceptability and recommendations for optimisation of the intervention (see online supplemental consent form: staff interviews).

Digitally recorded interviews and focus groups will be transcribed verbatim and anonymised. Transcripts will be analysed for patients and staff separately following Braun and Clarke's method for thematic analysis. Analysis will take an iterative approach, where data collection and analysis occurs concurrently, allowing the topic guide to be modified throughout to reflect emergent and/or priority themes.

## STATISTICS AND DATA COLLECTION
### Sample size calculation
As this is a feasibility study, no formal sample size calculation has been performed. An audit discussing sample size in pilot and feasibility studies concluded that while sample size justification is important, formal calculation may not be appropriate. The findings from the audit concluded that a median size of 30 in each arm is appropriate.[38] The study will aim to enrol 60 participants in the INT group over 1 year as this is a sufficient number to estimate a proportion of patients who have quit smoking by the day of surgery,[39] as well as to explore data collection processes, and inform sample size calculations for a potential larger trial. Recruitment of 60 participants in each group will allow us to measure recruitment and compliance rates with 95% CI width between 10% and 20%. It would also be enough to estimate the SD of questionnaires with reasonable accuracy for future planning of a larger trial.

### Data analysis
Appropriate summary statistics (eg, proportions and interquartile ranges, means and SDs) will be generated for the study feasibility and patient/clinical measures. Between-group measures (mean differences) will be generated alongside 95% CI; however, formal hypothesis testing will not be carried out as the aim here is not to conclusively prove efficacy and furthermore the size of sample is too small for any inferential tests to be meaningful. Participants will be kept in the groups they were allocated, regardless of compliance with treatment (intention-to-treat protecting against attrition bias). Analysis will be completed once all participants have completed all follow-up assessments. All data used in publication will be in an anonymous format in order to maintain patient study participation confidential.

## Handling missing data

A member of the research team will contact patients for any missing data (eg, questionnaires) via telephone and post. Where patients attend for follow-up clinic, the potential for missing data will again be limited. Imputation of missing responses is not proposed for patient-reported outcomes as this is not a definitive trial and no hypothesis testing will be performed.

## DATA MANAGEMENT AND QUALITY ASSURANCE
### Data management and confidentiality

Personal data will be collected from trial participants and hospital notes on CRFs, coded with the participant's unique trial number and initials. This will be held securely and strictly confidentially according to NHS policies. Patients in the semistructured qualitative interviews will be consented specifically for their name, address and contact telephone number to be shared with University of Birmingham (UoB) and University College London. Data will be transferred securely by encrypted end-to-end email and will not be labelled with private identifiable information. Interview response information will be kept encrypted on a computer in a locked office. No data that could be used to identify an individual will be published. Data will be stored on a secure server under the provisions of the Data Protection Act and/or applicable laws and regulations. Data may be accessed by external regulatory agencies and the Study Sponsor representatives and permission for this access will be documented within the participants consent form.

### Monitoring and audit

Onsite monitoring will be conducted as required by the UoB Clinical Research Compliance Team, with activities reported to the trials team and any issues noted followed-up to resolution. Additional onsite monitoring visits may be triggered, for example, by poor CRF return, poor data quality, low adverse event (AE) reporting rates, excessive number of participant withdrawals/deviations. Study data and evidence of monitoring and systems audits will be made available for inspection by the regulatory authority as required.

### Long-term storage of data

Trial data will be stored archived after the formal closure of the trial in accordance with archive policy and for the appropriate duration as per current legislation.

### Data access

On completion and publication of the study, individual participant data will be shared that underlie the results reported in study after deidentification. Additional related documents will be available including the study protocol, statistical analysis plan an analytical code. This data will be available in the beginning 3 months and 5 years following article publication to those who provide a methodologically sound proposal for analysis to achieve the aims in the approved proposal. All proposals should be directed to b.naidu@bham.ac.uk. To gain access, requestors will need to sign a data access agreement. Data will be available for 5 years at a third-party website.

## SPONSORSHIP AND INDEMNITY

The UoB will act as the sponsor for this study. Delegated responsibilities will be assigned to the Chief Investigator and the NHS Trusts involved in the study. The UoB has in force a Public Liability Policy and/or Clinical Trials policy, which provides cover for claims for 'negligent harm' and the activities here are included within that coverage.

## ETHICS AND DISSEMINATION

This study has obtained ethical approval from the NHS West Midlands Black Country Research Ethics Committee (Protocol V.3.0; REC number 19/WM/0097). This aim of this feasibility study is to inform a substantive trial. On completion, results will be published in a peer-review scientific journal.

**Acknowledgements** The Evolyst team for helping develop the Quit4Surgery web app.

**Collaborators** The Project MURRAY Investigators: Michael Shackcloth, Sarah Feeney, Lindsey Murphy, Magenta Black, Sridhar Rathinam, Rebecca Boyles, Syed Qadri, Karen Dobbs, Helen Shackleford, Zara Jalal, Christine Jordan, Christopher Golby, Ben Skirth.

**Contributors** All authors have made substantial contributions to the conception or design of the work, or acquisition, analysis or interpretation of the data. STL has participated in the study design, intervention development and drafted the manuscript. SK, AK and A-MB have participated in the intervention development. AF, OP, JB, RW, DRT and BN have participated in the study design, intervention development and have critically revised the manuscript. All authors have approved the final version.

**Funding** The research costs of the trial are funded by an investigator-initiated study grant from Johnson and Johnson (Grant #IIS-ENG-2017-01).

**Competing interests** None declared.

**Patient and public involvement** Patients and/or the public were involved in the design, or conduct, or reporting or dissemination plans of this research. Refer to the Trial design section for further details.

**Patient consent for publication** Not required.

**Provenance and peer review** Not commissioned; externally peer reviewed.

**ORCID iD**
Sebastian T Lugg http://orcid.org/0000-0002-7861-9108

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
