## [Reviewer comments · BMJ Open]

ARTICLE DETAILS

TITLE (PROVISIONAL)	Protocol for a feasibility study of smoking cessation in the surgical pathway before major lung surgery: Project MURRAY
AUTHORS	Lugg, Sebastian; Kerr, Amy; Kadiri, Salma; Budacan, Alina; Farley, Amanda; Perski, Olga; West, Robert; Brown, Jamie; Thickett, David; Naidu, Babu

VERSION 1 – REVIEW

REVIEWER	Giuseppe Gorini Oncologic network, prevention and research Institute (ISPRO), Florence, Italy
REVIEW RETURNED	06-Feb-2020

GENERAL COMMENTS	This paper reports an interesting protocol for a feasibility study of smoking cessation in the lung surgical pathway. The abstract and the full article are clear and well written. I only suggest to add two arrows in the Trial Schema (figure 1): the box "Patients declined participation in all aspects of study" (Box "Decline") needs to be linked to the other boxes through two arrows: one from the box "Patient approached at least 1 week prior to surgery discussed and information providerde" to the Box "Decline"; and the other arrow from the Box "Decline" to the Box "Patients are treated with the usual standard of care".
---

REVIEWER	Hiroshi Yokomichi University of Yamanashi
REVIEW RETURNED	23-Feb-2020

GENERAL COMMENTS	Protocol paper by Mister Babu Naidu et al. was about how feasible an RCT on smoking cessation support in thoracic surgical pathway. I would like to try to present comments for improving the protocol. [Major] 1. Inclusion criteria of patient age: I think that smoking rate would owe to sex and age generation. The researchers could consider stratification of the factors in enrollment or analysis.2. This pilot study is about feasibility. The researchers could gather more structured information about why people decline to take part in the study. This information would lessen the decline of the following RCT. The information would also contribute to the other study addressing smokers to quit the behaviour.3. Estimating smoking cessation rate would depend upon the characteristics of the studied population. From literature, could the researchers estimate the rate in the targeted population in the UK,
---

	please? This question would also be related to the rationale of this pilot study. [Minor] 4. Sample size calculation would also depend on the recruitment duration. Overall, I think that this pilot study would strengthen the future RCT and following other studies. This protocol would be well written, and be improved more.
--	---

REVIEWER	Matthew Steliga University of Arkansas for Medical Sciences USA
REVIEW RETURNED	30-May-2020

GENERAL COMMENTS	The authors share a well thought out manuscript detailing a study protocol for preoperative assessment and intervention related to preoperative smoking cessation. No concerns about this thorough protocol, and I would be very interested to see outcomes from future work from them. A couple thoughts when reading this which are not major points, but ideas to consider: The phrase "This should be offered weekly, preferably face-to-face, for a minimum of 4 weeks after the quit date" is an excellent, aggressive approach. In light of current limitations due to COVID, and some patients' difficulty with making face-to-face appointments for work or other reasons, specific consideration of back up plans such as- if patients are unable to come in for weekly visits, videoconferencing or telephone visits will be attempted. Is the Fagerstrom score standard practice at the clinic for all who smoke and gathered as part of their routine care, or only if enrolling in the study? I think it is only if enrolling in the study and consenting to data collection. Is it collected for those who refuse INT but accept UC? It would be very interesting to see if Fagerstrom score or other demographics such as income level, gender, age, etc could impact accepting intervention (INT) or refusing intervention and only accepting usual care (UC) and data collection. I understand in the feasibility you are looking at the number eligible for enrollment and the number accepting the study, but analyzing risk factors for refusal would be interesting, and perhaps allow future work to look at improving acceptance of iNT. We are interested ultimately in how all these variables affect cessation and duration of cessation, but the first step of accepting INT would be an interesting branch to analyze in the flow of the study. Perhaps this is your intent already, but i was not sure. Overall this study represents a great step in the process of integrating intervention in a particularly vulnerable high risk group with high smoking rates.
---

VERSION 1 – AUTHOR RESPONSE

Reviewer(s)' Comments to Author:

Reviewer: 1

Reviewer Name: Giuseppe Gorini

Institution and Country: Oncologic network, prevention and research Institute (ISPRO), Florence, Italy

Please state any competing interests or state 'None declared': None to declare

Please leave your comments for the authors below:

This paper reports an interesting protocol for a feasibility study of smoking cessation in the lung surgical pathway. The abstract and the full article are clear and well written.

Comment 1: I only suggest to add two arrows in the Trial Schema (figure 1): the box "Patients declined participation in all aspects of study" (Box "Decline") needs to be linked to the other boxes through two arrows: one from the box "Patient approached at least 1 week prior to surgery discussed and information provided" to the Box "Decline"; and the other arrow from the Box "Decline" to the Box "Patients are treated with the usual standard of care".

Response 1: We have amended the Trial Schema (figure 1) as suggested.

Reviewer: 2

Reviewer Name: Hiroshi Yokomichi

Institution and Country: University of Yamanashi

Please state any competing interests or state 'None declared': None declared.

Please leave your comments for the authors below:

Protocol paper by Mister Babu Naidu et al. was about how feasible an RCT on smoking cessation support in thoracic surgical pathway. I would like to try to present comments for improving the protocol.

[Major]

Comment 1: Inclusion criteria of patient age: I think that smoking rate would owe to sex and age generation. The researchers could consider stratification of the factors in enrolment or analysis.

Response 1: We thank the reviewer for this comment. This study is a feasibility study with primary aim 'To establish the number of patients who agree to participate in the intervention as a proportion of those eligible to enter the study' and therefore will not include stratification. The full scale RCT would most likely take the form of a step-wedge cluster randomisation so therefore would not include individual stratification for recruitment purposes but we would consider this for the analysis.

Comment 2: This pilot study is about feasibility. The researchers could gather more structured information about why people decline to take part in the study. This information would lessen the decline of the following RCT. The information would also contribute to the other study addressing smokers to quit the behaviour.

Response 3: We will also capture reasons for Failure of Recruitment as stated in the Protocol 'The proportion of patients that were missed, which should be minimal and proportion of patients who decline to take part will be recorded. Patients decline participation for many reasons, which should be captured whenever possible'.

Amended in Control Group of Usual Care 'As part of the feasibility study, for those patients who decline consent to receive the intervention, we ask if they would consent to be observed during the thoracic surgical pathway. Patients will also be invited for an optional telephone interview after discharge. This is to help understand smoking cessation rates in usual care as well as the reasoning for non-participation in the intervention'

Comment 3: Estimating smoking cessation rate would depend upon the characteristics of the studied population. From literature, could the researchers estimate the rate in the targeted population in the UK, please? This question would also be related to the rationale of this pilot study.

Response 3: 1 in 5 patients smoke before lung cancer surgery, with UK prevalence of smoking below 16%. Only 1 in 3 people report self-abstinence prior to lung cancer surgery, though with verification using CO this may be much lower. The use of objective verification and self-reported status in both intervention and usual care groups will further inform smoking cessation rates for the future trial.

[Minor]

Comment 4: Sample size calculation would also depend on the recruitment duration.

Response 4: We have factored in recruitment duration under Patients Recruited into Study which states 'Therefore the aim is to recruit 5 patients a month to each group over the 12 months recruitment period across all sites'.

Overall, I think that this pilot study would strengthen the future RCT and following other studies. This protocol would be well written, and be improved more.

Reviewer: 3

Reviewer Name: Matthew Steliga

Institution and Country:

University of Arkansas for Medical Sciences, USA

Please state any competing interests or state 'None declared': none

Please leave your comments for the authors below:

The authors share a well thought out manuscript detailing a study protocol for preoperative assessment and intervention related to preoperative smoking cessation. No concerns about this thorough protocol, and I would be very interested to see outcomes from future work from them. A couple thoughts when reading this which are not major points, but ideas to consider:

Comment 1: The phrase "This should be offered weekly, preferably face-to-face, for a minimum of 4 weeks after the quit date" is an excellent, aggressive approach. In light of current limitations due to COVID, and some patients' difficulty with making face-to-face appointments for work or other reasons, specific consideration of back up plans such as- if patients are unable to come in for weekly visits, videoconferencing or telephone visits will be attempted.

Response 1: Amended in Integration of full package of support into the surgical pathway (INT): Face-to-face interactions will occur during the surgical outpatient appointments, pre-clerking clinic and during hospital admission to reduce the number of additional visits. If patients are unable to attend face-to-face visits, videoconferencing or telephone visits will be attempted. Thus the INT will fit into the 'referral to treatment' target time frame and ensuring surgery is not delayed as recommended by NICE.

Comment 2: Is the Fagerstrom score standard practice at the clinic for all who smoke and gathered as part of their routine care, or only if enrolling in the study? I think it is only if enrolling in the study and consenting to data collection. Is it collected for those who refuse INT but accept UC? It would be very interesting to see if Fagerstrom score or other demographics such as income level, gender, age, etc could impact accepting intervention (INT) or refusing intervention and only accepting usual care (UC) and data collection. I understand in the feasibility you are looking at the number eligible for enrolment and the number accepting the study, but analysing risk factors for refusal would be interesting, and perhaps allow future work to look at improving acceptance of INT. We are interested ultimately in how all these variables affect cessation and duration of cessation, but the first step of accepting INT would be an interesting branch to analyse in the flow of the study. Perhaps this is your intent already, but I was not sure.

Response 2: The Fagerstrom score will be collected in both those consented to INT and to UC as per Table 1, which will help understand the nicotine dependence between groups. Baseline data including gender and age will also be collected in both groups as per Table 1.

We agree the importance of understanding refusal of participation, and therefore as part of the feasibility will assess reasons for refusal of study and intervention. See response to Reviewer 2 Comment 2.

Overall this study represents a great step in the process of integrating intervention in a particularly vulnerable high-risk group with high smoking rates.

FORMATTING AMENDMENTS (if any)

Required amendments will be listed here; please include these changes in your revised version:

Comment 1: Please provide figure legend/caption. Please include figure legends at the end of your main manuscript.

Response 1: We have provided figure legends as suggested.

Comment 2: Please re-upload your supplementary files in PDF format.

Response 2: We have up-loaded supplementary file in PDF as suggested.

VERSION 2 – REVIEW

REVIEWER	Giuseppe Gorini ISPRO, Florence, Italy
REVIEW RETURNED	03-Aug-2020

GENERAL COMMENTS	The Project Murray protocol is ready for publication.
---

REVIEWER	Hiroshi Yokomichi University of Yamanashi, Japan
REVIEW RETURNED	25-Jul-2020

GENERAL COMMENTS	I consider that the researchers have addressed all of my comments. I have no more concern. I appreciate their efforts to publish the important protocol.
--